# Adrenalectomy for Metastasis: The Impact of Primary Histology on Survival Outcome

**DOI:** 10.3390/cancers16040763

**Published:** 2024-02-13

**Authors:** Mariaconsiglia Ferriero, Andrea Iannuzzi, Alfredo Maria Bove, Gabriele Tuderti, Umberto Anceschi, Leonardo Misuraca, Aldo Brassetti, Riccardo Mastroianni, Salvatore Guaglianone, Costantino Leonardo, Rocco Papalia, Michele Gallucci, Giuseppe Simone

**Affiliations:** 1Department of Urology, IRCCS “Regina Elena” National Cancer Institute, 00144 Rome, Italy; alfredo.bove@yahoo.it (A.M.B.); gabriele.tuderti@ifo.it (G.T.); umberto.anceschi@ifo.it (U.A.); leonardo.misuraca@ifo.it (L.M.); aldo.brassetti@ifo.it (A.B.); riccardo.mastroianni@ifo.it (R.M.); salvatore.guaglianone@ifo.it (S.G.); costantino.leonardo@ifo.it (C.L.); michele.gallucci@ifo.it (M.G.); puldet@gmail.com (G.S.); 2Department of Urology, Fondazione Policlinico Universitario Campus Bio-Medico, 00128 Rome, Italy; andrea.iannuzzi@unicampus.it (A.I.); rocco.papalia@policlinicocampus.it (R.P.)

**Keywords:** adrenalectomy, adrenal metastasis, metastasectomy, primary histology, robotic surgery

## Abstract

**Simple Summary:**

The adrenal gland is often the site of metastasis coming from different primary tumors, including lung, kidney, breast, gastrointestinal cancer and melanoma. Metastasis-directed therapy is not yet well standardized and a multidisciplinary assessment is relevant for planning adequate management. Minimally invasive adrenalectomy is the most common treatment option, and many reports are available in the literature, demonstrating its feasibility. Previous studies showed increased survival in selected patients treated who had an adrenalectomy for metastasis, but all these findings remained inconsistent due to heterogeneous baseline clinical conditions, primary histology and tumor staging. Therefore, further investigations are needed to define the ideal candidate for adrenal metastasectomy in case of oligometastatic diseases. In the present report, we present the oncological outcome of a minimally invasive adrenalectomy for isolated adrenal metastasis, and the impact of primary histology on cancer-specific survival probability.

**Abstract:**

Adrenalectomy is commonly considered a curative treatment in case of adrenal gland as site of metastasis. In the present study, we evaluated the impact of primary tumor histology on survival outcomes after a minimally invasive adrenal mastectomy for a solitary metachronous metastasis. From May 2004 to August 2020, we prospectively collected data on minimally invasive adrenalectomies whose pathological examination showed a metastasis. All patients only received metastasectomies that were performed with curative intent, or to achieve non-evidence of disease status. Adjuvant systemic therapy was not administered in any case. Cancer-specific survival (CSS) was assessed using the Kaplan–Meier method. Univariable and multivariable Cox regression analyses were applied to identify independent predictors of CSS. Out of 235 laparoscopic and robotic adrenalectomies, the pathologic report showed metastases in 60 cases. The primary histologies included 36 (60%) renal cell carcinoma (RCC), 9 (15%) lung cancer, 6 (10%) colon cancer, 4 (6.7%) sarcoma, 3 (5%) melanoma and 2 (3.3%) bladder cancer. RCC displayed significantly longer survival rates with a 5-year CSS of 55.9%, versus 22.8% for other histologies (log-rank *p* = 0.01). At univariable analysis, disease-free interval (defined as the time from adrenalectomy to evidence of disease progression) < 12 months and histology were predictors of CSS (*p* = 0.003 and *p* < 0.001, respectively). At multivariable Cox analysis, the only independent predictor of CSS was primary tumor histology (*p* = 0.005); patients with adrenal metastasis from colon cancer and bladder cancer showed a 5.3- and 75.5-fold increased risk of cancer death, respectively, compared to patients who had RCC as primary tumor histology. Oncological outcomes of adrenal metastasectomies are strongly influenced by primary tumor histology. A proper discussion of the role of surgery in a multidisciplinary context could provide optimal treatment strategies.

## 1. Introduction

The adrenal gland is often a site of metastasis for different primary tumors, including lung, kidney, breast, gastrointestinal cancer and melanoma. Lung and breast cancers account for 39% and 35% of adrenal metastasis, respectively, while up to 40% to 50% of patients with late-stage melanomas and renal cell carcinoma (RCC) could experience metastasis in the adrenal gland [1].

Most adrenal metastatic tumors do not exhibit symptoms and only a limited number of reported cases have resulted in the development of Addison’s disease or hypoaldosteronism [2].

Reliable and sensitive methods of diagnostic imaging performed during follow-up protocols for treated cancer patients increased the probability of detecting oligo-progression and specifically adrenal metastasis [3]. The improved accuracy of these methods limited the rate of adrenal biopsies to selected scenarios, like multiple malignancies, the re-staging of a previous malignancy, determining the primary source in cases of unknown origin or distinguishing between benign and malignant adrenal masses when imaging findings are inconclusive [4].

Metastasis-directed treatment options include close observation, radiation therapy, local ablative therapy or surgery. Currently, there is a lack of randomized clinical trials providing evidence of a survival advantage of surgical metastasectomy over other treatment options. Choice among the alternatives could mainly depend on the clinical conditions of patients and the primary tumor staging [5].

Adrenal metastasis-directed therapy is not yet well standardized and a multidisciplinary assessment is relevant for planning adequate management. Surgical treatment of a solitary adrenal metastasis can be performed with an open, laparoscopic or robot-assisted approach. Minimally invasive adrenalectomy is the most common treatment option, whose feasibility and safety have been already demonstrated. Excellent outcomes including reduced hospitalization, decreased blood loss, and lower intraoperative and post-operative complication rate have been reported. Previous studies showed improved survival outcomes in selected patients treated with adrenalectomy for metastasis, but all these findings remained inconsistent due to heterogeneous baseline clinical conditions, primary histology and tumor staging [6,7,8].

Therefore, further studies are needed to define the ideal candidate for adrenal metastasectomy in oligometastatic disease.

In the present series, we report the oncological outcome of minimally invasive adrenalectomies for isolated adrenal metastasis and the impact of primary histology on cancer-specific survival probability.

## 2. Materials and Methods

### 2.1. Patient Selection

This study was approved by the internal Institutional Review Board Statement and Ethics Committee. All patients provided informed consent. From May 2004 to August 2020, 235 patients underwent laparoscopic or robotic adrenalectomy for any indication. From a review board-approved prospectively collected database, we selected all cases whose pathological examination showed a solitary or bilateral adrenal metastasis. Synchronous metastases were excluded. All metastasectomies were performed with curative intent or to achieve a non-evidence of disease (NED) status. Adjuvant systemic therapy was not administered to any patients. Preoperative workup included lab exams, a total body staging with CT or PET/CT scans and an endocrinologic counseling. All cases were discussed in our institutional multidisciplinary tumor board, which included an endocrinologist. Patients who received non-surgical therapy such as observation or stereotactic body radiation therapy (SRBT) were excluded. Baseline clinical, pathologic and follow-up data were prospectively collected. Preoperative adrenal biopsy was never performed.

### 2.2. Surgical Technique

#### 2.2.1. Robotic Approach

Two surgeons were involved in the cohort; two robotic platforms (da Vinci Si and Xi Surgical System; Intuitive Surgical, Sunnyvale, CA, USA) were used. Patients were placed in a mildly flexed extended flank position, and side-docking with transperitoneal five-port access was performed using a 30° scope. Patients, port placement and the instruments used meticulously reproduced partial nephrectomy setup [9]. The two 12 mm assistant ports allowed the introduction of one suction irrigation device and a 10 mm LigaSure device (Medtronic, Minneapolis, MN, USA).

On the right side, a straightforward approach to the right adrenal gland usually requires limited bowel mobilization. The triangular ligament was divided and the liver retracted superiorly by the bed assistant, providing wide exposure of the inferior vena cava and optimization of the surgical workspace. Gerota’s fascia was incised at the level of the upper pole of the kidney. The medial surface of the gland was bluntly dissected; occasionally identified small accessory veins were sealed as well as the adrenal main pedicle treated with 10 mm LigaSure.

On the left side, the splenic flexure was incised, and the splenorenal ligaments were divided. The spleen, bowel and pancreas tail were deflected medially. Gerota’s fascia was incised at the level of the upper pole of the kidney, and the adrenal gland was identified. The adrenalectomy was accomplished as previously described for the right side.

The specimen was removed in an Endocatch bag, and a drain was left in place [10].

#### 2.2.2. Laparoscopic Approach

All laparoscopic adrenalectomies in our study were performed using the transperitoneal approach, following the previously described technique [11].

### 2.3. Follow Up Schedule

Postoperatively, lab exams (including electrolytes, renal and liver function), urologic and endocrinologic visit, abdominal ultrasound and chest X-ray or CT scans were performed at six-month intervals during the first 2 years, followed by a yearly evaluation thereafter.

### 2.4. Statistical Analysis

The Kaplan–Meier method was used to evaluate cancer-specific survival (CSS). The whole cohort was split into two subgroups, RCC versus other primary histology, that were compared for the main clinical features. Chi-square test and Mann–Whitney test were applied to compare categorial and continuous variables, respectively. Age, gender, ASA Score, bilateral adrenal metastasis, tumor size, adrenal size, primary histology and disease-free intervals (DFIs) < 12 months were included in univariable and multivariable Cox regression analysis to identify independent predictors of CSS. A DFI was defined as the time from adrenalectomy to evidence of disease progression. RCC was the most common primary histology, therefore it was chosen as reference category for the Cox model. All *p* < 0.05 results were considered statistically significant. Statistical analysis was performed with the Statistical Package for Social Science (SPSS version 23, IBM, Chicago, IL, USA).

## 3. Results

Out of 235 laparoscopic and robotic adrenalectomies, pathological examination showed metastasis in 60 patients. The median age was 66 (IQR 58–72) years. The median hospital stay was 3 days (IQR 3–5). The high-grade complication rate (Clavien–Dindo classification 3–5) was 1.7%. Out of the 60 patients, one case of metastasis from RCC in the left adrenal gland experienced significant bleeding from a diaphragmatic vein surrounding the adrenal mass and required surgical revision. The median tumor size was 5.5 (IQR 3.5–7) cm. The primary histologies included 36 (60%) renal cell carcinoma (RCC 9 (15%) lung cancer, 6 (10%) colon cancer, 4 (6.7%) sarcoma, 3 (5%) melanoma and 2 (3.3%) bladder cancer. (Table 1). Soft tissue surgical margins were all negative.

The radiologic and macroscopic aspects of adrenal metastasis from the main primary tumors were reported in Figure 1.

Median follow-up was 34.5 months (IQR 15–75.2). The two cohorts, RCC and other primary histology were homogeneous for all the clinical variables except follow-up (*p* = 0.004) (Table 2).

Overall, 16 patients developed local or distant recurrence within 12 months from adrenal surgery (26.7%) and were treated with systemic therapies based on the primary tumor histology (chemotherapy, antiangiogenetic agents, tyrosine kinase inhibitors, immunotherapy). During the whole study period, 35 patients (58.3%) died of cancer; disease-specific mortality was 35% and 23.3% for the RCC and other histology groups, respectively.

Overall, 5-year progression-free survival, CSS and overall survival (OS) were 23%, 44.3 and 44.2%, respectively. One-year CSS were 97.2%, 66.7%, 50%, 50%, 85.7% and 50.4 for renal cancer, colon cancer, bladder cancer, sarcoma, lung cancer and melanoma, respectively (*p* < 0.001). (Appendix A).

Renal cell carcinoma displayed a more favorable cancer survival probability compared to the other histologies, with a 5-year CSS of 55.9% versus 22.8%, respectively (log rank *p* = 0.01; Figure 2).

At univariable analysis, a DFI < 12 of months and histology were predictors of CSS (*p* = 0.003 and *p* < 0.001, respectively). At multivariable Cox analysis, the only independent predictor of CSS was primary tumor histology (*p* = 0.005); patients with adrenal metastasis from colon cancer and bladder cancer had a 5.3- and 75.5-fold increased risk of cancer death, respectively, compared to patients who had RCC as their primary tumor histology (Table 3).

## 4. Discussion

In the past, the identification of adrenal metastases was predominantly performed during autopsy examinations. However, with the expanding employment of imaging techniques such as CT, MRI and PET in the diagnosis, staging, and follow up of malignancies, adrenal metastases are now often incidentally revealed early.

Recently, F-18 fluorodeoxyglucose positron emission tomography/computed tomography (18-FDG-PET/CT) has demonstrated its reliability with good sensitivity and specificity for the characterization of adrenal lesions [12]. Wu et al., in a recent review, reported a high accuracy of 18F-FDG-PET/CT in detecting adrenal metastasis in patients with lung cancer [13]. On the other hand, EAU guidelines do not recommend the use of a PET/CT scan for the diagnosis and staging of RCC, but it could play a role in restaging and follow up. Nonetheless, additional studies are needed to clarify its value [14].

In the present series, patients who developed adrenal metastasis, detected during the follow up of any other tumors, were referred to our department after discussion with a multidisciplinary team. However, many cases of adrenal metastasis derived from our institutional follow up for RCC. Specifically, about 25% of patients who underwent adrenalectomy had a diagnosis of an oligometastatic disease, and the most common primary histology was renal cell carcinoma (60%).

As far as minimally invasive adrenalectomy guarantees good perioperative outcomes compared to the open approach, there has been a growing acceptance of surgical treatment for patients with advanced localized disease or oligometastasis.

Some recent studies reported metastasectomy as an effective treatment option for RCC in selected patients. An observational study by Maisel et al. showed improved OS after metastasectomy in RCC, especially in the presence of a solitary metastasis and in some specific sites like the lungs and liver [15]. Therefore, Ishihara et al. reported a lower need of systemic therapy in patients who received a complete metastasectomy, compared to the ones treated with an incomplete resection, experiencing an improved quality of life [16].

Furthermore, in a recent series we highlighted the benefits of metastasectomy in terms of long-term OS probability for patients with metastatic renal cell carcinoma (mRCC) who received treatment at our tertiary referral center compared to those receiving systemic therapy alone [17].

However, concerning the topic of surgical treatment of mRCC, the only recommendation of the guidelines is to not offer tyrosine kinase inhibitor treatment to mRCC patients after metastasectomy and no evidence of disease [18].

In the present study, we reported oncological outcomes of patients treated with adrenalectomy for a solitary metastasis from different primary histologies.

Concerning adrenalectomy for a solitary metachronous metastasis in RCC, some reports confirmed improved survival outcomes [7,19,20]. In a recent retrospective multicentric series by Vlk et al. of 435 patients receiving adrenalectomy (195 of which were RCC as primary histology), median overall survival times of 37 (IQR 13–82), 23 (IQR 12–66) months and 16 (8–48) months were reported for RCC, other histologies and lung cancer, respectively, [6].

Similarly, in the present series, patients treated with minimally invasive adrenalectomy for RCC as a primary histology displayed a more favorable 5-year cancer-specific survival probability compared to all the other variants (55.9% vs. 22.8%, respectively, log rank *p* = 0.01).

Surgical treatment for adrenal gland metastasis from lung cancer is considered when the primary tumor can be managed through complete surgical resection or definitive radiation therapy or chemotherapy, and a NED status can be achieved. In the era of metastasis-directed therapy, local treatments have been increasingly used, providing benefits in terms of disease control, management of symptoms and possibly improved survival rates. Krumeich et al., in a retrospective cohort study, analyzed 122 patients with lung cancer who underwent adrenal metastasectomy. Median disease-free survival and OS time were 40 months and 47 months, respectively, with extra adrenal metastasis (HR: 3.52; *p* = 0.007) and small-cell histology (HR: 15; *p* = 0.04) having a negative impact on OS [21].

In our series, we reported 9 cases of adrenal metastasis from lung cancer with a 1-year CSS of 85.7% and a 5-year CSS of 26.8%. However, in a multivariable analysis, lung cancer as a primary histology did not impact cancer-specific mortality (HR 1.51, 95% CI 0.51–4.45, *p* = 0.456) after a minimally invasive adrenalectomy compared to the RCC variant histology.

In the present series, we also found three cases of adrenal metastasis from melanoma with a 1 yr CSS of 50.4%. The adrenal gland as site of recurrent melanoma can be a possible sanctuary exhibiting resistance to systemic treatments. Some studies showed the feasibility and the potential benefits of adrenal metastasectomy in carefully chosen patients [22,23]. Elliot et al. compared the outcomes between two groups: patients who received adrenalectomy for metastasis from melanoma and patients treated with systemic therapy alone. They reported an improved survival outcome of adrenalectomy compared to patients who did not receive surgery (116.9 vs. 11.0 months after adrenal metastasis diagnosis, respectively *p* < 0.001). Therefore, in the multivariable analysis, adrenalectomy was the strongest predictor of improved survival after adrenal metastasis diagnosis (HR: 0.27, 95% CI: [0.17–0.42]) [24].

Detection of isolated adrenal metastasis from colon cancer is quite rare and there is no consensus on management of this condition; however, available series showed favorable survival outcomes after adrenalectomy [6,7,25]. Vlk et al. reported an OS of 35 (12–83) months in patients with colorectal cancer after adrenalectomy, while Samsel et al. showed improved OS times for patients with adrenal metastases from colorectal cancer compared to the ones from non-small cell lung cancer (NSCLC) and from other histologies (29.5 vs. 10 vs. 10 months, respectively *p* = 0.012) [6,7]. Conversely, in our series, the six patients who received adrenalectomy for colon cancer metastasis displayed a 1-year and 2-year CSS of 66.7% and 33.3%, respectively, with an increased risk of cancer-specific mortality at multivariable analysis, compared to RCC (HR 5.33, 95% CI 1.85–15.34, *p* = 0.002).

Adrenal metastases from bladder cancer are less common and data concerning oncological outcomes are still lacking. Weiner et al., analyzing data from the National Cancer Database, found an improved OS for patients with metastatic bladder cancer who received metastasectomy at any site, compared to non-surgically treated patients (HR 0.85, 95%CI 0.79–0.91, *p* < 0.001) [26]. In the present series, we reported two cases of adrenal metastasectomy for bladder cancer that showed a 1-year CSS probability of 50%; at multivariable analysis, bladder cancer histology displayed a negative impact on survival outcomes compared to adrenalectomy for RCC (HR 75.49; 95% CI 5.90–965; *p* = 0.001).

Adrenalectomy is considered a relatively safe surgical procedure, but it can be associated with some complications including bleeding, lesions in close organs, tumor-spreading and infections. Vatansever et al. analyzed data from the European Surgical Registry EUROCRINE to compare clinical and surgical outcomes between laparoscopic and robotic adrenalectomy. Out of 1005 patients from 46 clinics that received a robotic or laparoscopic adrenalectomy, a lower complication rate (1.6% vs. 16.5%, *p* < 0.001), as well as a shorter hospital stay (length of stay ≤ 2 Days, 82.1% vs. 28.8%, *p* < 0.001), were found in a robotic rather than a laparoscopic cohort [27]. In our single-center experience, we showed that minimally invasive adrenalectomy is a feasible and safe technique, providing excellent perioperative outcomes. Only one case (1.7%) experienced a high-grade complication, which was significant post-operative bleeding requiring surgical revision.

Moreover, adrenal metastases could be treated with other non-surgical therapies, including stereotactic body radiation therapy (SBRT).

A recent meta-analysis and systematic review from 2009 to 2019, including 1006 patients treated with SBRT for adrenal metastasis, showed pooled 1- and 2-year rates of local control (LC) of 82% (95% CI, 74–88%) and 63% (95% CI, 50–74%), respectively, and a pooled 1- and 2-year OS rates of 66% (95% CI, 57–74%) and 42% (95% CI, 31–53%), respectively. Most patients had a primary lung malignancy (65.7%), concurrent chemo- or immunotherapy was rare and a sizeable minority of patients’ adrenal gland was the sole site of metastatic disease. Despite all the limitations and biases of the analysis, SBRT for adrenal metastases provided good 1-year LC with an excellent safety profile, and higher dose escalation may be associated with improved LC [28]. However, more reliable data are needed to confirm if the use of SBRT for adrenal metastasis confers a survival advantage. In the present series, adrenalectomy provided good 2-year CSS for RCC and acceptable CSS for lung cancer as the primary tumor histology (82.5% and 53.6%, respectively) compared to oncologic results reported for SBRT. Nonetheless, adrenalectomy after the failure of SBRT for adrenal metastasis may be a challenging procedure with an increased risk of vascular injuries.

Nowadays, many image-guided local ablative therapies, like radiofrequency (RF) ablation, cryoablation, chemical ablation and microwave ablation, have been applied to the treatment of adrenal metastases. Some studies reported that the RF ablation of adrenal metastases has shown favorable results in terms of local tumor control and recurrence-free survival. Nonetheless, these studies had a relatively small sample size and limited follow-up duration. Therefore, additional research is required to elucidate the true effectiveness of RF ablation in treating adrenal metastases [29,30].

We acknowledge some limitations of the present study. First of all, its single-center nature, as well as the low sample size can be considered the main drawbacks. Moreover, since the exact date of primary tumor diagnosis was not available for each case, the time to adrenal metastasis development was not included in the present analysis.

Finally, the heterogeneity of previous and subsequent oncological treatment lines, natural history and behavior of different primary tumors may impact on survival probabilities.

However, this study reported long-term oncological outcome of adrenalectomies performed for different primary tumors detected by an accurate follow up and a favorable 5-year CSS of patients with RCC compared to all the other histologies.

The role of surgery, as well as other metastasis-directed treatment options, should be properly discussed in multidisciplinary contexts to provide optimal disease control. Therefore, adrenal metastasectomy could be performed in selected cases of patients who are considered fit for surgery, displaying a favorable primary tumor histology and if a NED status can be achieved.

## 5. Conclusions

Oncological outcomes of adrenal metastasectomy is strongly driven by primary tumor histology. The prompt identification and treatment of solitary adrenal metastasis may lead to the achievement of a non-evidence of disease status, delaying the beginning of systemic treatments.

## Figures and Tables

**Figure 1 cancers-16-00763-f001:**
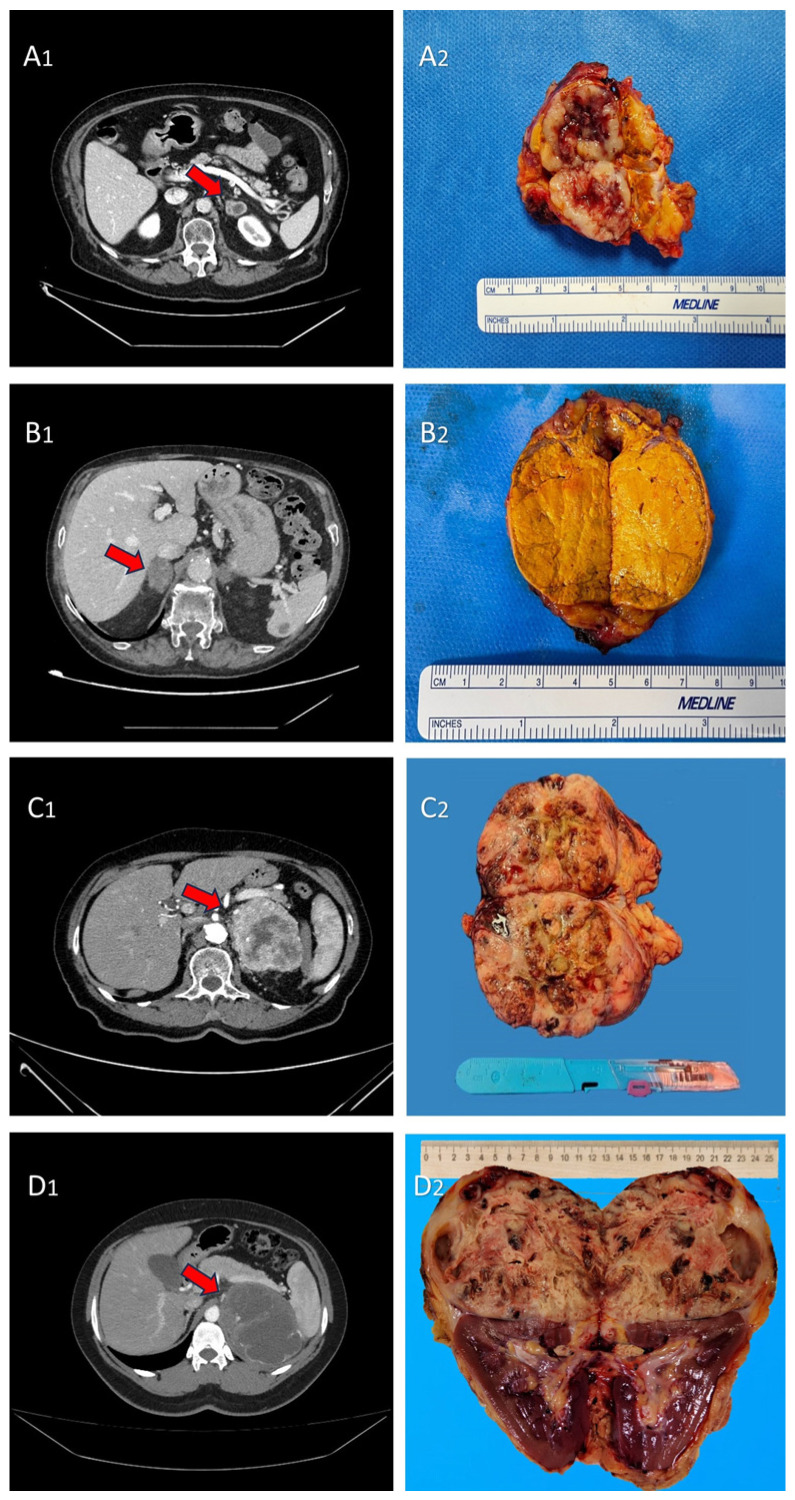
(**A_1_,A_2_**) CT scan image and macroscopic aspect of adrenal metastasis (red arrow) from non-small cell lung cancer (NSCLC); (**B_1_,B_2_**) CT scan image and macroscopic aspect of adrenal metastasis (red arrow) from urothelial carcinoma; (**C_1_,C_2_**) CT scan image and macroscopic aspect of adrenal metastasis (red arrow) from RCC; and (**D_1_,D_2_**) CT-scan image and macroscopic aspect of adrenal metastasis (red arrow) from melanoma.

**Figure 2 cancers-16-00763-f002:**
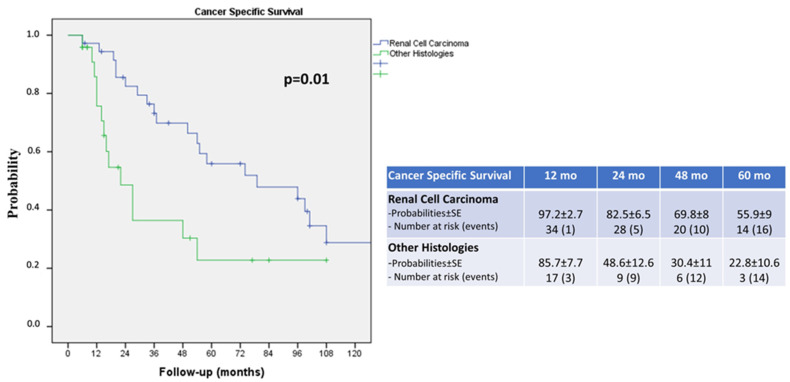
Kaplan–Meier curves showing the cancer-specific survival (CSS) of renal cell carcinoma (RCC) versus other histologies.

**Table 1 cancers-16-00763-t001:** Clinical and pathologic features of the study cohort.

DATA	Median (IQR) or *n* (%)
**Age (years)**	66 (58–72)
**Race**	
White Caucasian	60 (100)
**Gender**	
M	41 (68.3)
F	19 (31.7)
**Side**	
Right	22 (36.7)
Left	30 (50)
Bilateral	8 (13.3)
**ASA Score**	1–2 (43) 3 (17)
**Hospital Stay (days)**	3 (3–5)
**High Grade Complication Rate** **(Clavien–Dindo Classification 3–5)**	1 (1.7)
**Histology**	
Renal Cell Carcinoma	36 (60)
Lung Cancer	9 (15)
Bladder Cancer	2 (3.3)
Colon Cancer	6 (10)
Melanoma	3 (5)
Sarcoma	4 (6.7)
**Tumor size (cm)**	5.5 (3.5–7)
**Adrenal size (cm)**	6 (5–8)
**Follow-Up (mo)**	34.5 (15–75.2)

**Table 2 cancers-16-00763-t002:** Clinical and pathologic features of RCC (renal cell carcinoma) versus other histology subgroups.

	RCC (36)	Other Histology (24)	*p*
DATA	Median (IQR) or *n* (%)	Median (IQR) or *n* (%)	
**Age (years)**	63 (55–72)	68 (59–73)	
**Race**			
White Caucasian	36 (100)	24 (100)	1
**Gender**			0.42
M	25 (69.4)	14 (58.3)	
F	11 (30.6)	10 (41.7)	
**Side**			0.41
Right	11 (30.6)	11 (45.8)	
Left	19 (52.8)	11 (45.8)	
Bilateral	6 (16.7)	2 (8.3)	
**ASA Score**			0.57
**2**	26 (72.2)	17 (70.8)	
**3**	10 (27.8)	7 (29.2)	
**Hospital Stay (days)**	3 (3–3)	3 (3–3)	0.84
**High Grade Complication Rate** **(Clavien–Dindo Classification 3–5)**	1 (2.8)	0	0.6
**Tumor size (cm)**	5 (3.2–6.5)	5.5 (3.5–7)	0.3
**Adrenal size (cm)**	6 (5–7)	6 (3.7–9.5)	0.71
**Follow-Up (mo)**	54 (23–98)	15 (10–43)	**0.004**

**Table 3 cancers-16-00763-t003:** Univariable and multivariable regression analysis to identify predictors of cancer-specific survival.

	Univariable Analysis				Multivariable Analysis			
	HR	95% CI		*p*	HR	95% CI		*p*
**Age**	1.01	0.97	1.04	0.63	-	-	-	-
**Gender**	1.79	0.90	3.53	0.09	-	-	-	-
**Bilateral Adrenal Metastasis**	0.94	0.36	2.45	0.89	-	-	-	-
**Tumor Size**	0.98	0.85	1.14	0.85	-	-	-	-
**Adrenal Size**	0.88	0.70	1.12	0.89	-	-	-	-
**ASA Score (3 vs. 2)**	1.34	0.64	2.80	0.44	-	-	-	-
**Primary Histology** (RCC as ref category)				**<0.001**				**0.005**
**Colon**	6.94	2.56	18.81	**<0.001**	5.33	1.85	15.34	**0.002**
**Bladder**	1230.45	10.36	1470.56	**<0.001**	75.49	5.90	965	**0.001**
Sarcoma	0.94	0.12	7.06	0.95	1.04	0.13	7.9	0.97
Lung	1.51	0.51	4.45	0.46	1.52	0.51	4.51	0.45
Melanoma	1.41	0.18	10.69	0.74	1.49	0.19	11.28	0.70
**DFI < 12 mo**	3.05	1.46	6.33	**0.003**	1.89	0.80	4.46	0.15

## Data Availability

The data presented in this study are available in this article.

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
