# Peer review of "Adrenalectomy for Metastasis: The Impact of Primary Histology on Survival Outcome"

_cancers, 2024, doi:10.3390/cancers16040763_

Round 1

Reviewer 1 Report (New Reviewer)

Comments and Suggestions for Authors

Author Response

Major points:

  1. First of all, the reviewer is concerned if this study can add any clinical implications. The authors state ‘A proper discussion of the role of surgery in a multidisciplinary context could provide optimal treatment strategies’; however, it is unclear in which way the authors think this study would affect clinical practice for patients with adrenal metastases.

As well described by the reviewer, we reported safety and oncological outcomes of minimally invasive adrenalectomy for metastasis from different primary tumors. In view of these specific findings for each  primary tumor histology following, the role of adrenalectomy for should be discussed in a multidisciplinary team. If NED status can be achieved, surgical treatment of metastasis can be performed in case of RCC or lung cancer  and avoided in case of unfavorable histologies (like bladder or colon cancer). However,  we acknowledged several limitations of the study, as listed in the main text, above all we highlighted the need of  further studies.

  1. As related to the above question, primary sites per se have a strong impact on prognosis albeit unmodifiable (eg, metastatic urothelial carcinoma has dismal outcomes compared with metastatic RCC). Do the authors consider that it can be justified to omit adrenalectomy for such adverse cancers, or can it rather be encouraged to perform adrenalectomy for more favourable primary sites based on their findings and previous literatures?

As already explained in point 1, the primary histology, together with patient’s comorbidities could drive decision to perform adrenalectomy in case of a solitary metastasis at the adrenal gland in order to obtain a NED status and delaying systemic treatments. This comment has been added in discussion section.

  1. MATERIALS AND METHODS/RESULTS: Please indicate the number of cancer-specific deaths during the study period. The authors may use ‘RCC vs non-RCC’ for the category of primary histology in the Cox model(s) to avoid an unreliable estimate (eg, 95% CI of 5.9-965 for bladder cancer). If the number of events permits, the authors should include in the multivariable model all the variables that are associated with primary sites as well as baseline characteristics that differ across RCC vs non-RCC subgroups.

Along the whole study period, 35 patients (58.3%) died of cancer, 35% and 23.3% for RCC and non-RCC group respectively. This data has been added in the main text.

As suggested, clinical features of RCC and nonRCC subgroups have been compared (Table 2 has been added). Chi-square test and Mann-Whitney test were applied to compare categorial and continous variables, respectively. The two cohorts were comparable for all the variables except follow-up (p=0.004), therefore no further adjustments are needed in Cox  analysis.  However, we prefer to include all different primary tumor histologies (using RCC as reference category) in COX models in order to highlight the histology-specific risk of cancer specific mortality after adrenalectomy for metastasis. Besides, Cancer Specific Survival Probabilities of RCC group vs non RCC group have been already evaluated using The Kaplan Meier method.

  1. RESULTS: Please include the detail of high-grade complication in the Table 1 and/or in the Results section (it is now mentioned in the Discussion section only). Ideally, the primary disease characteristics and the culprit artery/vein for post-operative bleeding should be described.

We added the description of complication into results section. It was a case of metastasis from RCC at left adrenal gland and bleeding site was a diaphragmatic venous branch surrounding the adrenal mass.

Minor points:

  1. Do the authors have data on surgical margin? If so, please include as this is likely to affect the recurrence probability; otherwise, please state that in the limitations section.

In this series all surgical margins were negative. Results section has been updated.

  1. Typography of a duplication ‘..’ in the Line 96.

Thanks for suggestion, one dot has been deleted.

Reviewer 2 Report (New Reviewer)

Comments and Suggestions for Authors

Concerns:

-          It hasn't been a control group for comparison with other types of interventions. This prevents us from determining whether alternative therapeutic methods are more or less effective than adrenalectomy in terms of survival. For this reason is usefull to have a control group. If you are doing a comparative study, whic one is the standard of care?

-          I believe there might be a slight confusion in the term you mentioned “metastasectomies”. It's possible there might be a typographical error or a misunderstanding. If you are specifically discussing metastasis to the adrenal gland, it's helpful to clarify that in your context.

-          It is not entirely clear why only 60 out of the 235 patients were considered, given that the methods indicate all 235 patients had a lesion on the adrenal gland due to metastasis. If the analysis of post-operative histology is introduced into the study, it shifts from being a purely prospective study to a retrospective component as you are now looking back at data after the intervention has occurred.

-          In my opinion in the Kaplan-Meier curve, it is not the adrenalectomy that determines this curve, but rather the organ from which the metastases originate.

-          The limitations of this paper are somewhat concise. Firstly, it would be appropriate to establish a comparison group for the purpose of comparing various outcome types. Secondly, it is advisable to review both the results and the population under consideration. Additionally, clarification on whether this is a prospective or retrospective study is warranted, but overall it's a good report.

Comments on the Quality of English Language

minor english revision

Author Response

Reviewer 2

 It hasn't been a control group for comparison with other types of interventions. This prevents us from determining whether alternative therapeutic methods are more or less effective than adrenalectomy in terms of survival. For this reason is usefull to have a control group. If you are doing a comparative study, whic one is the standard of care?

      This is not a comparative study. The aim of our study was to report safety and oncological outcomes of minimally invasive adrenalectomy for metastasis from different primary tumors. According to our findings, adrenal metastasectomy could be perfomed in selected cases of patients fit for surgery, with favorable primary tumour histology and if a NED status can be achieved. Actually, we acknowledged that further studies are needed to define the standard of care.

-          I believe there might be a slight confusion in the term you mentioned “metastasectomies”. It's possible there might be a typographical error or a misunderstanding. If you are specifically discussing metastasis to the adrenal gland, it's helpful to clarify that in your context.

       In order to clarify, in discussion section we used the word “metastasectomy” meaning the resection of a metastasis at any site while we stated “adrenal metastasectomy” for the specific location at the adrenal gland.

-          It is not entirely clear why only 60 out of the 235 patients were considered, given that the methods indicate all 235 patients had a lesion on the adrenal gland due to metastasis. If the analysis of post-operative histology is introduced into the study, it shifts from being a purely prospective study to a retrospective component as you are now looking back at data after the intervention has occurred.

      From a review board approved prospectively collected database, out of 235 minimally invasive adrenalectomies performed for any indications, we selected 60 cases whose pathological examination showed a solitary or bilateral adrenal metastasis. The text has been modified in order to clarify patients selection. Synchronous metastases were excluded. Data collection and survival analysis have been prospectively perfomed.

-          In my opinion in the Kaplan-Meier curve, it is not the adrenalectomy that determines this curve, but rather the organ from which the metastases originate.

Thanks for the comment. In line with this idea, together with our findings, we reported that primary tumor histology is the independent predictor of Cancer specific survival probability.  The role of adrenalectomy for should be discussed in a multidisciplinary team. If NED status can be achieved, surgical treatment of metastasis can be performed in case of RCC or lung cancer  and avoided in case of unfavorable histologies (like bladder or colon cancer). However,  we acknowledged several limitations of the study, as listed in the main text, above all we highlighted the need of  further studies.

-          The limitations of this paper are somewhat concise. Firstly, it would be appropriate to establish a comparison group for the purpose of comparing various outcome types. Secondly, it is advisable to review both the results and the population under consideration. Additionally, clarification on whether this is a prospective or retrospective study is warranted, but overall it's a good report.

      The aim of the study is not comparative. However, the lack of a comparison with another group of patients not treated with adrenalectomy can be considered a limitation and has been listed in discussion section. Study population has been clearly described in materials and methods section. All data derived from a prospectively mantained database as well as the evaluation of outcomes.

Round 2

Reviewer 2 Report (New Reviewer)

Comments and Suggestions for Authors

no major comments 

Author Response

No Reply is needed

This manuscript is a resubmission of an earlier submission. The following is a list of the peer review reports and author responses from that submission.

Round 1

Reviewer 1 Report

Comments and Suggestions for Authors

This manuscript investigated the outcomes of patients treated with adrenalectomy for adrenal metastasis. From a sample of 235 patients who received robotic or laparoscopic adrenalectomy, the authors selected patients who presented with adrenal metastasis. They prospectively analyzed clinical data to determine the cancer specific survival (CSS) for different primary tumor histologies as well as assess independent predictors of CSS via univariate and multivariate cox regression. The CSS for renal cell carcinoma was lower than that for other histologies, suggesting a greater likelihood of cancer survival for RCC as compared to other primary tumors. The univariate regression implicated primary tumor histology and disease-free interval (DFI)<12 months as independent predictors of CSS. The multivariate analysis only identified histology as a predictor of CSS; it demonstrated that individuals with colon and bladder cancer showed 5.3- and 7.5-fold greater risks of cancer mortality, respectively. The article provides strong evidence that primary tumor histology is a determining factor in outcomes for adrenalectomy, but it should provide further justification/clarification for the comments below.

Specific comments: 

1)    In the discussion, the article discusses the CSS values for patients with RCC, lung cancer, melanoma, colon cancer, and bladder cancer. However, in the results section, it only reports the CSS for RCC individually, presenting the CSS for all other histologies as an aggregate data point. Please report the CSS values for each primary tumor histology in the results section.

2)    Please report whether the CSS values for each primary tumor histology correspond to the 5-year survival rates reported in the literature. 

3)    Is the timing to metastases available for each case? If yes, it would be interesting to include it in the univariable vs multivariable model analysis.

4)    What is the progression free survival and overall survival of the patients after adrenalectomy for adrenal metastasis?

5)    Please indicate why RCC primary tumor histology was chosen as the reference category in the univariable and multivariable analyses. 

6)    The discussion section describes survival benefits for adrenal metastasectomy in patients with melanoma and lung primary tumor histologies. However, the results of the study suggest that lung and melanoma histologies are not independent predictors of adrenalectomy outcomes. Please provide an explanation for this discrepancy. 

7)    The discussion states that complications arose in one patient to imply that minimally invasive adrenalectomy is a safe technique. Please provide an explanation for these complications and indicate why they would only occur in a small minority of patients. Please include the medical history of the patient in the manuscript and indicate whether similar complications are reported in the literature.

8)    Please report demographic information related to race for the sample cohort. 

9)    Please explain whether heterogeneity within the sample resulting from race differences may have influenced the results of the study.  

10) The right side of Figure 2 is cut off in the manuscript. Please expand the figure so all of the data is visible. 

Author Response

1)    In the discussion, the article discusses the CSS values for patients with RCC, lung cancer, melanoma, colon cancer, and bladder cancer. However, in the results section, it only reports the CSS for RCC individually, presenting the CSS for all other histologies as an aggregate data point. Please report the CSS values for each primary tumor histology in the results section.

- One-yr CSS were 66.7%, 50%, 50%, 85.7% and 50.4 for colon cancer, bladder cancer, sarcoma, lung and melanoma, respectively. The figure showing Kaplan-Meier curves for each primary histology was added as supplementary material. The text was modified accordingly.

2)    Please report whether the CSS values for each primary tumor histology correspond to the 5-year survival rates reported in the literature. 

- All survival outcomes for each primary histology were compared with available literature and commented in discussion section.

3)    Is the timing to metastases available for each case? If yes, it would be interesting to include it in the univariable vs multivariable model analysis.

- Thanks for the interesting suggestion, time to metastasis was not included in univariable and multivariable analysis since it was considered a co-linear variable of disease-free interval <12 months.

4)    What is the progression free survival and overall survival of the patients after adrenalectomy for adrenal metastasis?

- Five-yr progression free survival and OS were 23% and 44.2%, respectively. This data was added to the text.

5)    Please indicate why RCC primary tumor histology was chosen as the reference category in the Cox univariable and multivariable regression analyses. 

- In the present series RCC is the most common histology of adrenalectomy for metastasis, in addition a better survival outcome was observed at log rank analysis compared to the others. For these reasons RCC was chosen as reference category for univariable and multivariable regression analyses; the information was added in the materials and methods section.

6)    The discussion section describes survival benefits for adrenal metastasectomy in patients with melanoma and lung primary tumor histologies. However, the results of the study suggest that lung and melanoma histologies are not independent predictors of adrenalectomy outcomes. Please provide an explanation for this discrepancy. 

- This is not a discrepancy. Primary histology was included in univariable and multivariable regression analysis using RCC as reference category. We found that lung cancer and melanoma could benefit of adrenalectomy for a single metachronous metastasis as well as RCC, even if cancer specific survival could be influenced by subsequent available systemic therapies. Bladder cancer and colon cancer primary histology remained predictors of worse CSS probability compared to RCC after adrenal metastasectomy. Maybe this phenomenon could be due to a more aggressive behavior of these variant histologies.

7)    The discussion states that complications arose in one patient to imply that minimally invasive adrenalectomy is a safe technique. Please provide an explanation for these complications and indicate why they would only occur in a small minority of patients. Please include the medical history of the patient in the manuscript and indicate whether similar complications are reported in the literature.

- In the present study, only one case (1.7%) experienced a high-grade complication that was a significant post-operative bleeding requiring surgical revision. Medical history of this case was not relevant. Bleeding after adrenalectomy is commonly described in literature as a possible complication, however with the introduction of minimally invasive technique, the rate of significant bleeding as well as other complications became negligible.

8)    Please report demographic information related to race for the sample cohort. 

- The whole cohort is composed of white Caucasian patients, the feature has been added in Table 1.

9)    Please explain whether heterogeneity within the sample resulting from race differences may have influenced the results of the study.  

- The whole cohort is homogenous for race so that this feature cannot influence our results.

10) The right side of Figure 2 is cut off in the manuscript. Please expand the figure so all of the data is visible. 

- Fig. 2 has been replaced.

Reviewer 2 Report

Comments and Suggestions for Authors

Dear Author

The manuscript “Adrenalectomy for metastasis: the impact of primary histology on survival outcome” is an interesting one. 

1-      It has been presented in congress https://www.auajournals.org/doi/10.1097/JU.0000000000001975.10

2-      The written or verbal informed consent is needed. The ethical committee registration number of internal Institutional Review Board Statement and 83 Ethics Committee registration number is needed as well.

3-      In the manuscript cancer specific survival (CSS) has been considered. It would be better to report the overall survival (OS) as well.

4-      I am not a clinician so I recommend a clinician to judge the laparoscopic and robotic adrenalectomy process.

Author Response

1) It has been presented in congress https://www.auajournals.org/doi/10.1097/JU.0000000000001975.10

- Yes, It was presented in Podium Session at AUA 2021.

2) The written or verbal informed consent is needed. The ethical committee registration number of internal Institutional Review Board Statement and 83 Ethics Committee registration number is needed as well.

- These data have been reported at the end of the manuscript.

3) In the manuscript cancer specific survival (CSS) has been considered. It would be better to report the overall survival (OS) as well.

- These data have been included in Results section.

4) I am not a clinician so I recommend a clinician to judge the laparoscopic and robotic adrenalectomy process.

- Surgical technique has been described. Results are commented in discussion section

Round 2

Reviewer 1 Report

Comments and Suggestions for Authors

The revised manuscript by Ferriero et al has appropriately addressed the main concerns from the reviews and satisfactorily responded to other issues raised by the reviewer. The clarification and addition of new data have strengthened the overall manuscript.

Author Response

Reply to reviewers

Reviewer 1:

1) In the discussion, the article discusses the CSS values for patients with RCC, lung cancer, melanoma, colon cancer, and bladder cancer. However, in the results section, it only reports the CSS for RCC individually, presenting the CSS for all other histologies as an aggregate data point. Please report the CSS values for each primary tumor histology in the results section. - One-yr CSS were 66.7%, 50%, 50%, 85.7% and 50.4 for colon cancer, bladder cancer, sarcoma, lung and melanoma, respectively. The figure showing Kaplan-Meier curves for each primary histology was added as supplementary material. The text was modified accordingly.

2) Please report whether the CSS values for each primary tumor histology correspond to the 5-year survival rates reported in the literature. - All survival outcomes for each primary histology were compared with available literature and commented in discussion section.

3) Is the timing to metastases available for each case? If yes, it would be interesting to include it in the univariable vs multivariable model analysis. - Thanks for the interesting suggestion, time to metastasis was not included in univariable and multivariable analysis since it was considered a co-linear variable of disease-free interval <12 months.

4) What is the progression free survival and overall survival of the patients after adrenalectomy for adrenal metastasis? - Five-yr progression free survival and OS were 23% and 44.2%, respectively. This data was added to the text.

5) Please indicate why RCC primary tumor histology was chosen as the reference category in the Cox univariable and multivariable regression analyses. - In the present series RCC is the most common histology of adrenalectomy for metastasis, in addition a better survival outcome was observed at log rank analysis compared to the others. For these reasons RCC was chosen as reference category for univariable and multivariable regression analyses; the information was added in the materials and methods section.

6) The discussion section describes survival benefits for adrenal metastasectomy in patients with melanoma and lung primary tumor histologies. However, the results of the study suggest that lung and melanoma histologies are not independent predictors of adrenalectomy outcomes. Please provide an explanation for this discrepancy. - This is not a discrepancy. Primary histology was included in univariable and multivariable regression analysis using RCC as reference category. We found that lung cancer and melanoma could benefit of adrenalectomy for a single metachronous metastasis as well as RCC, even if cancer specific survival could be influenced by subsequent available systemic therapies. Bladder cancer and colon cancer primary histology
remained predictors of worse CSS probability compared to RCC after adrenal metastasectomy. Maybe this phenomenon could be due to a more aggressive behavior of these variant histologies.

7) The discussion states that complications arose in one patient to imply that minimally invasive adrenalectomy is a safe technique. Please provide an explanation for these complications and indicate why they would only occur in a small minority of patients. Please include the medical history of the patient in the manuscript and indicate whether similar complications are reported in the literature. - In the present study, only one case (1.7%) experienced a high-grade complication that was a significant post-operative bleeding requiring surgical revision. Medical history of this case was not relevant. Bleeding after adrenalectomy is commonly described in literature as a possible complication, however with the introduction of minimally invasive technique, the rate of significant bleeding as well as other complications became negligible.

8) Please report demographic information related to race for the sample cohort. - The whole cohort is composed of white Caucasian patients, the feature has been added in Table 1.

9) Please explain whether heterogeneity within the sample resulting from race differences may have influenced the results of the study. - The whole cohort is homogenous for race so that this feature cannot influence our results.

10) The right side of Figure 2 is cut off in the manuscript. Please expand the figure so all of the data is visible.

- Fig. 2 has been replaced.